# Chitin and Chitosans: Characteristics, Eco-Friendly Processes, and Applications in Cosmetic Science

**DOI:** 10.3390/md17060369

**Published:** 2019-06-21

**Authors:** Cristina Casadidio, Dolores Vargas Peregrina, Maria Rosa Gigliobianco, Siyuan Deng, Roberta Censi, Piera Di Martino

**Affiliations:** School of Pharmacy, University of Camerino, 62032 Camerino, Italy; cristina.casadidio@unicam.it (C.C.); dolores.vargas@unicam.it (D.V.P.); maria.gigliobianco@unicam.it (M.R.G.); siyuan.deng@unicam.it (S.D.); roberta.censi@unicam.it (R.C.)

**Keywords:** Chitin, chitosan, cosmetics, biodegradability, biomaterials, polysaccharides, green technology, marine cosmetic ingredients, marine green source, marine resources

## Abstract

Huge amounts of chitin and chitosans can be found in the biosphere as important constituents of the exoskeleton of many organisms and as waste by worldwide seafood companies. Presently, politicians, environmentalists, and industrialists encourage the use of these marine polysaccharides as a renewable source developed by alternative eco-friendly processes, especially in the production of regular cosmetics. The aim of this review is to outline the physicochemical and biological properties and the different bioextraction methods of chitin and chitosan sources, focusing on enzymatic deproteinization, bacteria fermentation, and enzymatic deacetylation methods. Thanks to their biodegradability, non-toxicity, biocompatibility, and bioactivity, the applications of these marine polymers are widely used in the contemporary manufacturing of biomedical and pharmaceutical products. In the end, advanced cosmetics based on chitin and chitosans are presented, analyzing different therapeutic aspects regarding skin, hair, nail, and oral care. The innovative formulations described can be considered excellent candidates for the prevention and treatment of several diseases associated with different body anatomical sectors.

## 1. Introduction

Global warming, waste disposal, air pollution, and natural resources exhaustion have been recognized as the dominant cause of many environmental disasters. For these problems, politicians, environmentalists, and industrialists are encouraging the use of organic products able to respect our habitat. Presently, organic products are increasing their marketing power and social interest thanks to the high expectations of consumers towards ambient and human health [1,2]. These goods, to be considered “green” or “organic”, must be produced with renewable energy, possess characteristics such as fast degradation, minimize waste production, and be environmentally friendly [3]. “Green Chemistry” and its principles allow the adoption of technologies that fully meet the requirements of cost, safety, and performance. Anastas and Eghbali [4,5] listed the 12 principles of green chemistry, cornerstones for design, development, and industrial assessment of green products, achieving sustainability at the molecular level. Recently, Cervellon et al. [6] outlined a series of indispensable concepts for a product that aims to be classified as "green", especially in the world of cosmetics:

**Biodegradable:** if the goods are disposed of in nature, their degradation process under the action of microorganisms (fungi, bacteria, and molds) will be faster than a conventional product;

**Biodynamic:** more natural and sustainable production processes favor the formation of a product (philosophy of taking/giving back to nature); 

**Ecological:** the product has respect for the environment by limiting damage, without evaluating the mode of production processes;

**Natural:** found in nature without chemicals or human transformations (it is not certified by organizations); 

**Organic:** products and raw materials must be grown without the use of pesticides, synthetic fertilizers, genetically modified organisms, or ionizing radiation. They also should come from animals that have not taken growth hormones or antibiotics.

A strategy that favors sustainable consumption, minimizing ambient impact, is to raise awareness of purchasing products based on biopolymers from renewable resources [7,8]. Among biopolymers that are biodegradable and made from renewable raw materials, chitin and chitosan are widely used in many sectors such as biomedical, biotechnology, water treatment, food, agriculture, veterinary, and cosmetics [9]. These biopolymers are polymers produced by living organisms such as plants, microorganisms, and animals, and it is no coincidence that chitin and chitosan are considered in recent publications as polymers of the future, thanks to their innumerable properties and numerous advantages in their use [3,10,11,12]. Specifically, in the 21st century, there has been an increase in the production and use of regular cosmetics, because they are perceived by the consumers to possess numerous benefits such as limited environmental incidence, and being beneficial for the skin [13].

In this review, the general properties of chitin and chitosan as natural marine polysaccharides will be described, highlighting their innovative eco-friendly extraction processes and industrial and biomedical applications. In the second part, the development of chitin, chitosans, and their derivates as cosmetic products will be investigated considering the definition of cosmetic product by the European Union Cosmetic Regulation. Cosmetics are any products, substances, or preparations other than drugs intended to be applied on the external surfaces of the human body (epidermis, hair system, nails, lips, external genital organs) or on teeth and mucous membranes of the mouth for the exclusive or prevalent purpose of cleaning them, perfuming them, modifying their appearance, correcting body odors, protecting, or keeping them in good condition [14]. Starting from this regulation, we also provide examples of cosmetics with drug-like substances, up to now without specific legislation, considering them as promising starting points for new cosmetic developments.

## 2. Chitin and Chitosans: Structure, Properties, and Applications 

Biopolymers are used in widespread sectors of applied scientific areas such as biomedical, food, chemical, and industrial fields [15,16,17]. Among biopolymers derived from natural sources, polysaccharides have gained great attention thanks to their peculiar biomedical and physicochemical properties such as biodegradability, biocompatibility, non-toxicity, renewability, and ready availability. They are preferred compared to synthetic polymers for their low price and high presence in natural living organisms [11,18]. Polysaccharides can be classified by their nature in acid (carrageenan, alginic acid, hyaluronic acid, chondroitin sulfate), basic (chitin and chitosan, polylisine) or neutral (dextran, agarose, pullulan) [19]. In terms of basic polysaccharides, chitin and its major derivative product chitosan are the most important and abundant marine polymers in the world, and their physicochemical properties depend on their origin and extraction method. 

### 2.1. Physical and Chemical Characterization of Chitin and Chitosans

Over 200 years ago, the French botanist Henri Braconnot, thanks to his research on edible mushrooms, discovered a new polysaccharide that in 1823 took the name of chitin (Figure 1a) [20]. Chitin is a natural polysaccharide derived from numerous living organisms and is the second most abundant polymerized carbon present in nature, after cellulose. Because of the similar chemical backbone, chitin is often wrongly attributed to cellulose, but instead of a hydroxyl group, chitin presents acetamide groups (CH_3_CONH_2_) at C2 position [21]. Its crystalline structure is composed of poly (1,4)-linked *N*-acetyl-2-amino-2-deoxy-β-D-glucose (GlcNAc) with some residues of 2-amino-2-deoxy-β-D-glucose and presents white, nitrogenous, inelastic, and hard material features. The physical and chemical properties of chitin, and chitosan, depend on their different raw materials and method of preparation. In nature, there are three chitin allomorphs with crystalline forms—α (the most common), β, and γ—which can be characterized by different instruments (X-ray Diffraction (XRD), Solid-State Nuclear Magnetic Resonance (SSNMR) Spectroscopy, Infrared (IR) Spectroscopy, Solid-State Cross-Polarization/Magic-Angle-Spinning (CP–MAS) ^13^C NMR Spectroscopy, and Thermogravimetric Analysis (TGA)). The α-chitins are mainly sourced in fungi, yeasts, krill, lobsters, crabs, shrimps, and insects; β-chitins in squid pens, while γ-chitins in *Ptinus* beetles and Loligo squids [22]. The principal difference among these allomorphs mostly depends on their structure: α-chitin shows an orthorhombic unit with chains arranged in antiparallel sheets or stacks; β-chitin has a monoclinic unit with parallel chains; γ-chitin report a unit cell with a random chain trend predominating (two up one down) [23,24,25]. Thanks to its antiparallel microfibril orientation, α-chitin owns strong inter- and intramolecular bonds that prevent diffusion of small molecules into the crystalline phase and is the preferred chitin allomorph for the industrial uses. Regarding β-chitin parallel chains, the intramolecular hydrogen bonds have weak strength and, for this reason, solubility, but also reactivity and the swelling can be managed [26]. 

In 1859, Rouget observed the mechanism of chitin partial *N*-deacetylation, which leads to the formation of its derived compound: chitosan (Figure 1b) [27]. The reaction consists of the protonation of the chitin amino group at C-2 position of glucosamine. Specifically, when the chitin degree of deacetylation reaches approximately 50%, chitin conformation becomes soluble in acid water solution and changes to chitosan cationic structure [24]. Chitosan is a linear heteropolysaccharide consisting of poly (1,4)-linked 2-amino-*β*-D-glucose (GlcN), which can be in solid semicrystalline form or in solution, and represents the progenitor of the numerous chitin families deacetylated to different degrees [28]. The properties of soluble chitosan refer to the degree of deacetylation, molecular weight, distribution of remainder acetyl groups, ionic concentration, pH, isolation, and drying conditions. Chitosan represents a collective name for the *N*-deacetylated chitin derivates family with different degrees of deacetylation, and is prepared via three different procedures. One is the (partial) thermochemical deacetylation of chitin in the solid state under basic conditions (NaOH), the second is enzymatic hydrolysis in the presence of a chitin deacetylase (chitin + H_2_O ⇌ chitosan + acetate), and chitosan can be found also in nature in the structural component of some fungi [29,30].

#### 2.1.1. Degree of Deacetylation

The degree of deacetylation (DD) is the percentage of glucosamine (C_6_H_13_NO_5_) monomers present in the chitin structure. DD produces a huge effect on the solubility of chitin, unlike chitosan, due to its inability to dissolve in aqueous acid solution such as acetic acid. When the DD turns over 50%, chitin becomes soluble in acid-diluted solution and changes to the chitosan cationic structure [24]. The determination of DD can be performed by numerous techniques such as Ultraviolet (UV) Spectroscopy, IR Spectroscopy, Proton and Carbon Nuclear Magnetic Resonance Spectroscopy (^1^H and ^13^C NMR), SSNMR Spectroscopy, Gel Permeation Chromatography (GPC), Circular Dichroism, Residual Salicylaldehyde Analysis, Titration Methods, Elemental Analysis, High-Performance Liquid Chromatography (HPLC), Thermal Analysis, and Mass Spectrometry (MS) [31]. Heidari et al. provided several characterizations of DD with Fourier Transform Infrared Spetroscopy (FTIR) following Sabins’s law [32]: Absorbance ratio = (A)_amide_/(A)_hydroxyl_(DD) = 97.67 − (26.486 (A_1655_/A_3450_))

They obtained for chitin a DD of 50.1% while for three different natural chitosan the DD were 73.5%, 82.3%, and 82.5%, respectively [33]. In another work, DD was evaluated with a new FT-Raman approach identifying the experimental bands of chitin and chitosan based on density functional theory (DFT) quantum chemical calculations [34]. Dimzon et al. for DD determination used an IR absorbance ratio method improved with the use of partial least squares (PLS). The IR spectral region was settled from 1500 to 1800cm^−1^ and the results obtained were equal to those obtained with potentiometric titration and better than those obtained with the IR absorbance ratio conventional method [35].

#### 2.1.2. Molecular Weight

The molecular weight (MW) of chitin depends on its origin source and is related to the method of decalcification with HCl, achieving the maximal depolymerization. The MW can be determined by HPLC or by viscometry, where the intrinsic viscosity (η) was determined in 0.1 M acid acetic and 0.2 M sodium chloride solution, following the Mark–Houwink equation [29]:(η) = k⋅M^α^ = 1.81 × 10^−3^⋅M^0.93^

The weight-average MW of chitin is reported to range from 0.4 to 2.5 × 10^6^ [36,37].

Chitosan MW compared to the chitin one is lower, due to the N-deacetylation. Chitosan MW depends on the degree of polymerization (DP), where oligomers with DP of 8 or less are water-soluble irrespective of deacetylation conditions such as time, temperature, concentration of sodium hydroxide, and pH value [28,31]. The weight-average MW of chitosan is from 1 × 10^5^ to 5 × 10^5^ Da and can be determined by several methods, including: light-scattering spectrophotometry, GPC, and viscometer [31]. In recent years, size-exclusion chromatography (SEC) has been another approach adopted to better determinate the MW. Kang et al. coupled SEC with multi-angle static laser light scattering (MALLS) to better improve the characterization of chitosan MW [38]. In another work, Weinhold and Thöming connected SEC instruments with a triple detector array including in-series right and low-angle light-scattering (RALS-LALS) and viscometer [39].

#### 2.1.3. Solubility

Chitin shows variations in its solubility according to different sources. The biopolymer is insoluble in all the usual solvents such as neutral water, salt solutions, and most organic solvents, but presents solubility with other solvents such as hexafluoroacetone sesquihydrate, hexafluoroisopropanol, chloroalcohols (with sulfuric acid), water mineral solutions, and in a mixture of dimethylacetamide with 5% of lithium chloride. Its poor solubility is due to the presence of highly hydrophobic properties and its extensive semicrystalline structure [36]. Chitin is a polysaccharide with intra- and intermolecular hydrogen bonds, which makes it difficult to dissolve in the previous indicated solvents.

Chitosan is insoluble in neutral water and its solubility is settled with acid solutions such as lactic, acetic, glutamic, and hydrochloric acid solutions (pH up to 6.5), due to the lower number of *N*-acetylated groups and due to its primary amino groups (with a pKa of 6.3) which become protonated, leading to a positively charged polymer and giving the characteristics of a strong base. However, when the pH reaches the value of 6.0 (and above), the polysaccharide becomes insoluble and precipitates due to the deprotonation of the amines. Presently, purified chitosans with high DD are commercially available in a broad range of MW, both in the form of base and as a salt readily soluble in water without the use of acid solutions [40]. Commonly, the solubility of chitosan decreases when pH raises from physiological to basic values, and with increase the ionic strength (salting-out effect) or the MW [24]. There are other determining factors that have important effects on chitosan solubility: temperature, average of DD, and DP. The most common solvents for the solubilization of chitosan are: acetic acid (1% with pH close to 4); formic acid (0.2–100%); 1% hydrochloric acid; lactic acid; and diluted nitric acid. Recent research found a neutral chitosan solution with the use of glycerol 2-phospate as solvent [24]. On the other hand, chitosan is insoluble in sulfuric and phosphoric acid [29].

#### 2.1.4. Derivatives

Several chitin derivatives are described in the literature, among which chitosan appears to be the main product. Chitosan can be obtained by chitin following two different processes: the enzymatic hydrolysis or via chemical deacetylation, as shown in Figure 2. In addition to this, numerous other chitin products have been identified [24]. Thanks to the availability of the amino group, chitin can be associated with macromolecules such as proteins, carotenoids, and glucans [28].

Regarding chitosan, its structure has three active groups which can be chemically modified to change specific properties and activities. These groups are: the primary (C-6) and secondary (C-3) hydroxyl groups at the level of which non-specific reactions occur, and the amino group (C-2), where specific reactions can be distinguished. Solubility, and physical and mechanical properties of chitosan can be altered with the chemical modification of these reactive groups, attributing to its new derivatives further properties. The main reactions involving the C-3 and C-6 positions are esterification and etherification, while for the amino-C-2 position the quaternization of the amino group is carried out. In C-2, where the aldehyde function reacts with -NH_2_ by reductive amination, a reaction can be settled in aqueous solution under mild conditions favoring the introduction of different functional groups on chitosan using acrylic reagents in an aqueous medium [31].

### 2.2. Chitin and Chitosans Biological Properties

Both chitin and chitosan exhibit various biological properties such as: anticholesterolemic, wound-healing agents, anticancer, fungistatic, hemostatic, analgesic, antiacid, antiulcer, immunoadjuvant, etc. [19,21,29,31,40,41,42,43,44]. In cosmetic science, chitin and chitosan have been investigated as potential excipients and as biological active agents, thanks to their peculiar properties such as no toxicity, biocompatibility, and biodegradability. This review will discuss four of the main characteristics and qualities that make these polysaccharides excellent candidates in the formulation of care products: antimicrobial and antioxidant activities, mucoadhesive and penetration properties. The indispensable physicochemical parameters of the polysaccharides that promote the biological effect are summarized in Table 1.

#### 2.2.1. Antimicrobial Activity

Chitin and chitosans have shown great antimicrobial activity against a large sector of microorganisms such as bacteria, fungi, and yeast. The mechanism behind its antibacterial and antifungal activity is still unknown but different hypotheses have been theorized in this regard. One of these causes can be associated with the impermeable coat formation due to crosslinking between the polycation nature of the polysaccharides and the negatively charged cell surface at pH lower than 6.5. This layer would prevent the intake of nutritional substances into the bacterial cells, leading to microorganism death. The other mechanism involves the chelating agent properties of chitin and chitosan and their influence on organism growth. The third procedure concerns the easy permeation of low-MW chitosan through cell wall bacteria and its association with deoxyribonucleic acid (DNA) and the suppression of ribonucleic acid (RNA) and protein synthesis. Together with low MW, also high-level of DD enhance the antibacterial activity of chitosan with an improvement of permeabilizing effect and a better electrostatic binding to the bacteria membrane [45,46,47,48]. Liu et al. have recently developed a bioactive natural preservative material for cosmetic formulation based on kojic acid (KA), a natural pyrone compound, and chitosan oligosaccharides (COS), following a one-step environmentally friendly approach. They investigated the antibacterial and antifungal activities against two fungi and three Gram-negative and three Gram-positive bacterial strains. The results showed that with an increase in degrees of substitution of COS with KA (owing to positively charged grafting groups), an enhanced antimicrobial activity of the system can be observed [49]. Another interesting candidate of natural cosmetic preservative was provided by Juliano and Magrini. The synergistic activity of chitosan and methylglyoxal, a compound of manuka honey, was tested against different Gram-positive and Gram-negative bacteria and several strains of *Candida*, achieving an improvement of antimicrobial activity efficiency [50].

#### 2.2.2. Antioxidant Activity

The antioxidant activity of chitin, chitosan, and derivatives corresponds to their scavenging ability against different oxygen radical species such as alkyl, superoxide, hydroxyl, and DPPH (2,2-diphenyl-1-picrylhydrazyl). The mechanism is still unclear but should be related to the chelation of free metal ions by the polysaccharide hydroxyl and amino groups, which leads to the formation of a stable system. The in vitro tests highlighted that high percentages of DD simultaneously with a low MW favors a more efficient scavenging action [51,52,53]. Zhang et al. have provided a good example of chitosan derivative as a potential source of antioxidants for cosmetic applications. They synthetized three different combinations of N,N,N-trimethyl chitosan salts with acetylsalicylate (TMCSAc), ascorbate (TMCSAs), citrate (TMCSCi), and gallate (TMCSGa) following the ion-exchange method. Their results displayed an inhibition of free radical chain reaction due to the synergistic action of the acid anion and the trimethyl chitosan cation. Therefore, TMCSAs and TMCSGa products showed better antioxidant activity [54]. Chitosan could also be used to produce a liposomal delivery system for antiaging cosmetic formulations. In a recent work, a chitosan-coated liposome was proposed for the controlled release of coenzyme Q10 and alpha-lipoic acid, using the Cell Counting Kit-8 (CCK8) colorimetric assay to evaluate the antioxidant activity and cytotoxicity of the formulations. The results revealed that chitosan-liposome system has low cytotoxicity with an excellent antioxidant activity (clearing reactive oxygen species (ROS) from H_2_O_2_) [55].

#### 2.2.3. Mucoadhesive Properties

The main component of mucus is mucin, a glycoprotein rich in negative charges that interact with the positive ones of chitosan. Previous studies have established that the physical and chemical characteristics of the chitosan favor an improvement of the mucoadhesive properties related prevalently to DD and MW. Unlike what happens for antioxidant and antimicrobial activities, in this case there is an improvement of the mucoadhesion when polysaccharides are used with a high degree of DD and high MW [56,57]. Pereira et al. designed *Aloe vera*/vitamin E/chitosan microparticles for burn treatment application that could also be used in the future as a cosmetic proposal. They performed in vitro mucoadhesion test demonstrating and confirming the adhesive property of the system correlated with the presence of chitosan [58].

#### 2.2.4. Penetration Enhancement Properties

The permeation enhancement carried out by chitosan is associated with the opening and destruction of epithelial tight junctions by a decrease in transepithelial electric resistance. The chemical nature of the mechanism is based on the electrical interaction between the positive charges of chitosan and the cell membrane, leading to a re-association of the proteins associated with the tight junctions [59,60,61]. In 2016, researchers developed cationic and anionic acrylic nanocapsules with a diameter of 150 nm, embedded into chitosan gel for cosmetic application. The chitosan cationic charged surface allowed a deeper skin penetration of acrylic capsules by induction of the tight junctions opening in the stratum granulosum, below the stratum corneum [62]. Kojima et al. investigated the distribution of chitosan, using a hair cosmetic ingredient to improve the texture of hair surface, via time-of-flight secondary ion mass spectrometry (TOF-SIMS). They analyzed the penetration of chitosan as a hair conditioning agent and the results showed how cationic chitosan equally incorporated on the hair surface, highlighting an important difference between virgin and bleached hair. The amount of cationic chitosan adsorbed on the virgin hair was lower than the bleached hair, because bleaching leads to a negative charge enhancement due to cysteic acid group formation [63].

### 2.3. Chitin and Chitosans General Applications

The applications of chitin and chitosan include uses in a variety of areas, such as food industry, wastewater treatment, agriculture, cosmetics, pharmaceutical, and medical applications, paper production, and textiles. The wide world of chitin and chitosans applications is shown in Table 2.

## 3. Extraction of Chitin and Chitosans from Natural Sources

Chitin and chitosan are considered important marine renewable sources due to their high availability as garbage from the seafood processing industry; chitin availability was estimated to be approximately over 10 billion tons annually [29,140]. Presently, most producers for commercial purposes of chitin and chitosan are in Poland, India, Norway, Australia, USA, and Japan [141]. There are mainly two chitin extraction methods conducted in the industry—chemical or biological. Both extraction strategies of chitin consist of two phases—deproteinization with alkaline treatment at high temperatures, and demineralization with dilute hydrochloric acid. The sequence of these two phases is interchangeable depending on the source and the proposed use of chitin. The third phase mainly depends on the starting waste material: if the chitin is extracted from squid pens, a final non-pigmented white powder is obtained. On the other hand, chitin powder isolated from crustacean sources assumes a pale pink color, thus necessitating the bleaching process, which requires the use of hydrogen peroxide, oxalic acid, or potassium permanganate [45,142]. Figure 2 provides a scheme of chitin preparation from marine shell waste following the chemical process. These synthetic methods are very risky and have many disadvantages due to high temperature and high concentration of acid and alkali solutions. In addition, the production of chitin and chitosan by chemical process has different industrial drawbacks such as: high energy consumption, long handling times, greater solvent wasted, high environmental pollution, high production of waste, and difficulty in recovering waste products such as pigment and proteins [143,144,145]. As an alternative to the chemical process, bioextraction of chitin has been studied as a newly green ecological process.

### 3.1. Bioextraction of Chitin

Chitin is a constituent of the organic matrix of different marine organisms, including: arthropod exoskeletons such as crustaceans (crab, shrimp, lobster, krill, crayfish, barnacles) and insects (cockroach, beetle, true fly and worm); mollusk endoskeletons; fungi (*Aspergillus niger, Mucor rouxii, Penicillum notatum*); yeasts; algae; cuttlefishes; and squid pen. Crustacean shell is composed of 30–40% proteins, 30–50% mineral salts (principally calcium carbonate and phosphate), and 13–42% of chitin with its different chemical structures—α-, β-, γ-form. In minimal percentages, carotenoids (mainly astaxanthin and its esters) and lipids from visceral or muscular residues can also be found in shellfish waste [146]. Chitin is extracted from crustacean shell waste with three different steps: demineralization, deproteinization, and bleaching/decoloration. Chemical demineralization and deproteinization present several issues that prevent optimal control of reactions: depolymerization, anomerization, and decrease of MW by altering the properties of purified chitin. To solve these problems the application of bioextraction process is preferred, with two different methods: the employment of proteolytic enzymes to digest proteins, or the microorganism-mediated fermentation (Figure 3). The limitations of these biological procedures are high cost, lower yield, and the final properties of the products [143,147,148].

#### 3.1.1. Chitin Enzymatic Deproteinization

One of the proposed biological alternatives is the use of proteases for deproteinization of crustacean shells, mainly deriving from plant, microbial, and animal sources. This method avoids alkaline treatments and produces, in addition to chitin, protein hydrolysates with nutritional value. Depending on the starting waste, the protease can lead to various deproteinization yields according to the conditions tested. The demineralization should be performed first, to increase the tissue permeability, decrease the presence of potential enzyme inhibitors, and promote the action of the proteolytic enzyme. Chymotrypsin, Papain, Trypsin, Alkalase, Devolvase, Pepsin, and Pancreatin are the major proteolytic enzymes used to extract and separate the protein and chitin residues from shrimp waste. The final products present more beneficial physicochemical properties compared to the ones obtained following other methods. Proteases can be purified and extracted commercially at high cost and less efficacy, and crudely extracted, mainly derived from bacteria but also from fish viscera [149,150]. The crude extracted proteases are cheaper and more efficient thanks to the presence of coexisting proteases. Mhamdi et al. in their study reported the evaluation, characterization, and application of thermostable serine alkaline proteases from actinomycete strain *Micromonospora chaiyaphumensis* S103 for chitin extraction from shrimp shell (*Penaeus kerathurus*) waste powder. The percentage of deproteinization obtained after 3 hours of hydrolysis at 45 °C and pH 8.0 with an enzyme/substrate (E/S) ratio of 20 U/mg reached 93%, one of the best results in the literature compared to the use of other proteases [151]. In another work, crude digestive alkaline proteases from the viscera of *Portunus segnis* proved to be very efficient in the production of chitin by deproteinization of blue crab (*P. segnis*) and shrimp (*P. kerathurus*). In this case, the percentage of deproteinization achieved was near 85% for blue crab shells and 91% for shrimp shells with an E/S ratio of 5 U/mg of proteins after 3 h incubation at 50 °C [152].

#### 3.1.2. Chitin Bacteria Fermentation

Another inexpensive approach for chitin extraction from seafood wastes consists of two different methods of fermentation, with and without lactic acid bacteria (LAB). Fermentation can be carried out by adding selected strains of microorganisms, following one-stage and two-stage fermentation, co-fermentation/subsequent fermentation, or from endogenous microorganisms (auto-fermentation). 

##### Lactic Acid Bacteria Fermentation

LAB fermentation has been studied as an innovative method for enzymatic extraction of chitin and can be combined with chemical treatments reducing the amount of acid and alkali needed [148]. The ability of LAB strains is to ferment waste materials and simultaneously produce organic acids in situ (lactic and acetic acids). During fermentation, two fractions are obtained—a liquid fraction rich in proteins, minerals, and pigments and a solid phase containing crude chitin (which can be separated by filtration and washed with water). The separation into two fractions occurs thanks to the action of lactic acid that promotes the precipitation of the chitin and the production of calcium lactate after reaction with calcium carbonate. The lactic acid, obtained by conversion of glucose, at the same time promotes the lowering of the pH and consequently the activation of proteases. This methodology has also been used for the recovery of other products from silage shrimp waste, such carotenoids. The most used bacterial strains for fermentation are *Lactobacillus* sp. strain especially *L. plantarum*, *L. paracasei* and *L. helveticus*. Recently, Castro et al. have extracted and purified chitin from *Allopetrolisthes punctatus* crabs using *Lactobacillus plantarum* sp.47, a Gram-positive bacterium isolated from Coho salmon that produce high lactic acid concentrations. They obtained a 99.6% demineralization, 95.3% deproteinization and 17 mg of lactic acid/g silage choosing optimal fermentation parameters (60 h fermentation, 10% inoculum, 15% sucrose and 85% crab biomass) [153].

##### Non-Lactic Acid Bacteria Fermentation

For chitin recovery with non-lactic acid bacteria crustacean shell fermentation bacteria and fungi were used as the inoculum source: *Pseudomonas* sp., *Bacillus* sp. and *Aspergillus* sp. In their research, Ghorbel-Bellaaj et al. isolated a protease bacterium identified as *Pseudomonas aeruginosa* A2. By evaluation of protease activities and spectral analysis, they showed how chitin extracted by the biological method was similar to commercial α-chitin. The ability to deproteinize shrimp waste to produce chitin was also highlighted, overcoming the disadvantages of chemical deproteinization [154]. In another work, the same researchers followed a Plackett–Bhenken design to better improve deproteinization and demineralization efficiencies of shrimp shells with *P. aeruginosa* A2. They used these optimized variables: shrimp shell concentration (50 g/L), glucose concentration (50 g/L), incubation time (5 days), and inoculum size (0.05 OD), obtaining a deproteinization of 89% and demineralization of 96% [155]. 

Most commercial bacterial proteases are mainly produced by *Bacillus* sp., and Hajji et al. have extracted from the waste of crab shells chitin and fermented-crab supernatants after fermentation using six different strains of *Bacillus*. Using specific assays, it has been discovered that fermented-crab supernatants possess interesting antioxidant and antibacterial properties [156].

Regarding fungi as an inoculum source, three different proteolytic strains of *Aspergillus niger*, namely 0576, 0307, and 0474, were selected by Teng et al. thanks to their protease activity necessary to produce chitin. The aim of their study was to obtain two distinct sources of chitin by adding the fermented shrimp shells to mushrooms directly. The proteolytic enzymes released by fungi during the deproteinization and demineralization of the shrimp shell lead to the release of amino acids, which is a source of nitrogen to promote the growth of fungi [157].

### 3.2. Enzymatic Deacetylation of Chitin

Chitosan can be derived from chitin by chemical or enzymatic deacetylations. Chemical deacetylation is usually preferred because it is cheaper and guarantees suitability for mass production, but, at the same time, presents disadvantages such as energy consumption and increased environmental pollution due to the alkaline conditions. To overcome these drawbacks, an innovative enzymatic method that exploits chitin deacetylases has been explored: enzymatic deacetylation of chitin. Hembach et al., with their research conducted in 2017, have chosen fungal, viral, and bacterial chitin deacetylases, producing 14 possible partially acetylated chitosan tetramers with a defined degree of acetylation and a pattern of acetylation, also showing a purification method [158]. 

## 4. Applications in Cosmetics

Among polysaccharides, chitin, chitosan, and their derivatives offer intangible qualities related to antiaging, matrix metalloproteinase (MMP) inhibitor, antioxidant, and antifungal properties. The use of chitin and chitosans has been suggested in different fields of skin, oral, nail, and hair care applications, obtaining formulations able to treat diseases related to teeth, hair, nails, and skin. 

### 4.1. Skin Care Applications

Chitin, chitosan, and their derivatives are widely used in cosmetics especially because they exert antioxidant, cleansing, protecting, humectant, and antioxidant functions. Most of the chitin and chitosan cosmetic applications described in the following paragraphs refer mainly to: antiaging and moisturizing agents, ultraviolet protective compounds, skin cleansing products, and boosting factors of skin-essential functions such as protection, absorption, thermal regulation, defense, reservation, and synthesis.

#### 4.1.1. Antiaging and Moisturizing Agent

Skin aging is commonly a consequence of the intrinsic aging that occurs with the progression of years, but also of extrinsic aging caused by external factors as cigarettes, UV radiation, air pollution, etc. Dryness, relaxation, roughness, and skin tissue laxity are the main characteristics of skin aging and its association with UV ray exposure represents one of the most documented causes of hyperpigmentation origin and wrinkles, leading to the phenomenon known as a photoaging [159]. Recently, researchers have shown how chitosan, especially of high MW, possesses film-forming properties that can promote a reduction in cutaneous water loss and an increase in skin elasticity and smoothness, making it interesting in moisturizing cosmetic applications [160]. Transparency of film is one of the desirable features due to its great impact on the cosmetic fields, especially the ability to obtain a product with no further visible changes. Han and Floros calculated the transparency in the following equation [161]:
Transparency = A_600_/x
where x is the film thickness (mm) and A_600_ is the absorbance at 600 nm. The greater transparency value is represented by lower transparency. In 2019, Montenegro and Freier patented a transparent tissue dressing material based on deacetylated native chitosan suitable for different cosmetic applications such as peelings and face masks [162]. In another work, the skin treated with chitosan film neutralized in citrate buffer (with or without hyaluronic acid) prepared by Libio et al., demonstrated a desquamation of the stratum corneum and a significant increase in the degree of hydration within 10 minutes on a model of pig skin compared to skin without treatment. These results suggest that the biocompatible film, in the absence of glycerol, promotes a cosmetic effect regarding skin exfoliation, thanks to the bioadhesive properties of chitosan [163].

Morganti et al. in a study conducted in 2013 showed the antiaging activity of a particular cosmetic formulation based on the use of chitin nanofibril-hyaluronan (CN-HA) block copolymeric nanoparticles. These CN-HA block-copolymers should be used with invasive and non-invasive therapies in aesthetic medicine, exploiting drug-delivery properties. In vitro studies have highlighted the ability to easily encapsulate different active compounds (such as lutein) and in vivo studies have demonstrated the innovative antiaging properties of these formulations, with encouraging long-lasting results [164,165,166]. As an alternative to the common chemical polymers, Rajashree and Rose investigated the antiaging power of a gel based on collagen, chitosan, and *Aloe vera.* Chitosan improved the stability and the capability to induce local cell proliferation, and in combination with collagen enhanced the skin fibroblast biocompatibility, attachment, and proliferation. The final system was able to increase the rejuvenation and regeneration of the skin [167].

#### 4.1.2. Ultraviolet Protective Cosmetics 

Chitin and chitosan show adhesive properties (thanks to the electrostatic interactions between the positively charged polysaccharides and negatively charged keratin-based structures [168]), water resistance, cytocompatibility, and UV absorption (below 400 nm), qualities necessary for the formulation of protective creams. The main radiation characteristics of solar rays are UV, specifically UV-A (320–400 nm), and UV-B (290–320 nm). These radiations cause numerous adverse reactions on the skin such as sunburn, skin degeneration, photosensitivity, phototoxicity, photoaging, immunosuppression, and skin cancer. To prevent such diseases caused by excessive exposure of the skin to solar radiation, chitosan-containing sunscreens with substances with strong protective efficacy are used [159,169]. Ito et al. used urocanic acid (UCA), the major UV-absorbing chromophore in the skin, to prepare nanofibrils of urocanic acid-chitin by UCA hydrolysis. They examined the protective effect of the formulation against UV-B radiation demonstrating in vivo their protective effect but also the ability to inhibit erythema induced by UV-B irradiation and solarization cell generation [170]. In a recent work, solar emulsion based on chitosan nanoparticles (150–500 nm) were prepared with annatto, ultrafiltered annatto, saffron, and ultrafiltered saffron. All formulations were synthesized via ionotropic gelation and showed good preservation and low toxicity, while minimal sun protection was observed with sun protection factor (SPF) values ranging from 2.15 to 4.85. The storage stability was evaluated, and the final system showed a good storage (regarding pH and viscosity) at room temperature for up to 90 days [171]. 

#### 4.1.3. Skin Cleansing 

Cleaning the skin means removing from its surface contaminating foreign substances that are acquired during a simple air exposure or cosmetic product application. Chitosan and its derivatives, thanks to their cationic nature, can be used as positively charged vehicles in the delivery of products for personal cleaning. They can indeed exploit the ionic attraction between their charge and the anionic nature of the surface of the skin [172]. A liquid cleansing composition with moisturizing and exfoliating dual properties was designed and patented by Massaro et al. To date, they have not used chitin as a starting ingredient in the specific cosmetic cleansing but it is listed among the next substances suitable for the formulation of a new cleansing product [173].

### 4.2. Nail Care Applications 

The nail is a structure produced by the skin and is therefore an appendage of the skin itself. Nail disease, onychosis, has a distinct classification with respect to skin diseases. Onychomycosis, a fungal infection of the nail unit, is a common disorder that is currently treated with broad-spectrum antimycotics delivered through topical administration and/or in combination with systemic oral drugs. Most cases of onychomycosis are caused by skin infections or because of a nail trauma (mechanical trauma or exposure of chemical agents), altering the natural barrier function of the nail. A topical agent such as nail lacquer represents a valid topical formulation to prevent fungal infections compared to creams and solutions, since it favors a better stay of the formulation at the site of action. Hydroxypropyl chitosan (HPCH), a semisynthetic derivative of chitosan, has proved to be a valid candidate for the delivery of active products to nails, acting as a protective film that preserves nail structure, protecting keratin, maintaining hydration with a decrease of dystrophy signs in psoriatic nails [174,175]. Two recent studies conducted by Cantoresi et al. regarding the use of HPCH have shown the efficacy of the treatment of dystrophy in psoriatic nails by the association of the derivative chitosan with horsetail extract (*Equisetum arvense*) and methylsulphonyl-methane (DMSO_2_). During the preliminary study, the efficacy of this formulation was identified and then confirmed by a secondary and randomized placebo-controlled double-blind trial. This research covered a period of 24 weeks where the synthesized product proved to have been statistically superior to placebo [174,176]. Ghannoum et al. exemplify the effects of a HPCH-based nail solution compared to urea and isopropyl alcohol effects on the bovine hoof structure. They used HPCH as a starting material for the composition of film able to prevent fungal infection (not to treat it) and to protect keratin and maintain hydration with restructuration of nails. Unlike the chemicals normally used in cosmetic treatments (isopropyl alcohol or urea), repeated application of the HPCH nail solution can prevent the occurrence of new or recurrent fungal infections by increasing hardness, tensile strength, and flexural strength of the hoof sample compared to the untreated control. HPCH also reduces the crumbling area of the sample after abrasion and penetration of dermatophyte hyphae [177].

### 4.3. Hair Care Applications

Hair is the piliferous ends that grow at the level of the skin and are made of solid proteins, in a high percentage keratin, composed of numerous amino acids among which are lysine and cysteine, but also from melanin, which gives color to hair. Different factors cause damage to hair such as the use of high temperatures in the drying phase (hairdryer, curling tongs, or hair straighteners), during the coloring phase with contact to aggressive chemical agents, and exposure to UV rays or contact with chlorine. Here, as with previous applications, the use of chitin and chitosan demonstrate peculiar characteristics: electrostatic interaction (with negatively charged hair), hydrophobicity (removing oils and sebum from hairs), antibacterial and antifungal activities, interaction with hair keratin creating a transparent and elastic film at the level of the hair surface, favoring and increasing softness and strength of the hair. Due to these properties, chitin and chitosan are used in cosmetic formulations such as hair tonics, hair colorants, hair sprays, permanent wave agent, rinses, hair gels, etc. As an example of a drug product developed for scalp treatment, we report a study of Matos et al., who developed a delivery system based on chitosan nanoparticles (about 236 nm) loaded with minoxidil sulfate (MXS-NP) in a 1:1 weight ratio, for targeted release to hair follicles. Chitosan nanoparticles were obtained using low-MW chitosan and tripolyphosphate as crosslinked agent. MXS-NP were able to accumulate in the hair follicles and support the release of drugs more than twice compared to the previous microparticles loaded with MXS alone, maintaining relevant therapeutic concentrations for over 12 hours. The loading of MXS into chitosan nanoparticles proves to be a promising strategy for the release of drugs to hair follicles, improving the topical treatment of alopecia [178]. Nonetheless, the same strategy could be investigated for the delivery of active natural compounds in cosmetic formulations. In 2017, researchers studied the properties of hair covered with thin films consisting of collagen, chitosan, and hyaluronic acid mixture, evaluating their respective surface and mechanical properties. Chitosan and collagen were mixed in different volumetric ratios: 25:75, 50:50, and 75:25, while percentages of 1, 2, and 3% of hyaluronic acid were added to the final solution. The film was obtained through solvent evaporation method at room temperature. Thin film formulation brings numerous benefits to hair such as thickness increase, favorable mechanical properties, better appearance, and conditioning [179]. 

### 4.4. Oral Care Applications

When referring to dental care, we mean the organs within the oral cavity such as the teeth and the gum, which is a soft connective tissue that surrounds the teeth and covers the alveolar process. Among various dental diseases, we can include anodontia (genetic disorder of the congenital absence of teeth), dental caries (degenerative disease of tooth tissues), tooth ware (loss of dental substance by means other than dental caries or dental trauma), periodontal disease (inflammation of dental tissue), and bruxism (rubbing of teeth during sleep). As far as gingiva-related diseases are concerned, gingivitis (an increase in the thickness of the free gum) and periodontitis (an infection involving tooth-support tissues leading to loss of gingival attachment) are often frequent. Chitosan and its derivatives are used in the treatment of oral problems through the formulation of gels, dentifrices, sprays, chewing gum, mouthwashes and microspheres, to prevent diseases such as oral mucositis, plaque formation, periodontal problems, and bacterial growth control. 

#### 4.4.1. Caries Treatment

Dental caries, one of the most common oral health problems worldwide, is a pathological process caused by the organic acids produced by the dental plaque biofilms present on the enamel surface. The formation of dental caries and enamel surface white spot lesions are dynamic processes associated with an imbalance between demineralization and remineralization, which must be treated with a remineralizing agent. He et al. have recently published an anti-cariogenic system able to prevent and treat early caries and white lesions and to promote remineralization. A mineral solution of nanocomplexes of carboxymethyl chitosan/amorphous calcium phosphate (CMC/ACP) was previously characterized and its antibacterial activity was evaluated on enamel-coated blocks of saliva. The results shown an inhibition of adherence of *Streptococcus mutans* and *Streptococcus gordonii* for 90% and 86% respectively, and a reduction of biofilm formation about 45% and 44%. Additionally, CMC/ACP reduced the attachment of *Fusobacterium nucleatum* (promoter of biofilm development) to streptococcal biofilm by 75% and acting on both zeta potential of the bacterial suspension and cytochrome c-binding bacteria [180]. Regarding the formulation of toothpastes, Achmad et al. synthesized a chitosan-based dentifrice (5%) from white shrimp (*Litopenaeusvannamei*) able to reduce the number of colonies of *Streptococcus Mutans* in the case of early childhood caries. The effectiveness of chitosan toothpaste was more efficient than chitosan with 2.5% toothpaste and placebo toothpaste [181].

#### 4.4.2. Erosive Tooth Ware Treatment

The loss of dental substance caused by chemical and mechanical processes that do not involve bacteria belong to erosive tooth ware. In a study published in 2018, researchers tested the preventive erosive effect of toothpastes in permanent teeth and, for the first time, in deciduous teeth. They noted that the deciduous teeth had a lower initial superficial microhardness than the permanent teeth. No significant differences were observed between the two types of teeth when fluoride toothpaste (four different formulations) were used but, in the treatment with placebo dentifrice without fluoride, the deciduous teeth showed a significantly greater softness compared to the permanent teeth. The presence of chitosan as a thickener in anti-erosion toothpaste AmF-NaF-SnCl_2_ allowed a better preventive effect only for deciduous teeth while the NaF anti-erosion children’s toothpaste formulation favored better efficacy for both types of teeth [182]. Beltramea et al. evaluated in vitro anti-erosive effects of phosphorylated chitosan solutions in bovine dentin. The loss of dentin surface, the surface hardness and the modulus of elasticity were measured by profilometry, nano-hardness, and scanning electron microscopy (SEM), demonstrating a preventive and therapeutic action of chitosan in the treatment of dental erosion [183].

#### 4.4.3. Gingivitis Treatment

An oral disease of lichen planus is the desquamative gingivitis, where the gingiva becomes inflamed and swollen, and takes on a more reddish color. Desquamative gingivitis can be treated with administration of topic corticosteroids, such as hydrocortisone sodium succinate, a synthetic water-soluble derivative of hydrocortisone, with peculiar properties: antivirus, anti-coma, and anti-inflammatory [184]. Last year, Davoudi et al. developed via and environmentally friendly process a chitosan/gelatin/keratin composite containing hydrocortisone sodium succinate as a buccal mucoadhesive patch to treat desquamative gingivitis with pH values suitable for the oral cavity [185].

#### 4.4.4. Periodontitis Treatment

Periodontitis, initiated by bacteria accumulation, consists of the destruction of dental structures (loss of alveolar bone, periodontal ligament tearing) that can lead to the actual loss of teeth [186]. The disease, associated with an inflammatory state, can be treated with the administration of statin drugs such as atorvastatin. To increase the efficacy of the drug and improve its in situ administration, Özdoğan et al. have developed a bioadhesive delivery system based on chitosan, for the local administration of atorvastatin. Good viscosity and bioadhesive properties have been found, favoring an easy applicability at the level of the periodontal pocket and a sustained release of the drug. In vitro studies using human gingival fibroblast cells showed that cytokine release decreased with atorvastatin and the presence of chitosan enhanced anti-inflammatory activity. Following the administration of the system they found, compared to the control, a dental bone healing and a decreased level of proinflammatory cytokines such as interleukin-1beta (IL-1β), IL-6; IL-8; IL-10, and anti-inflammatory transforming growth factor (TGF) as TGF-β1, TGF-β2 and TGF-β3 [187]. The same research group in another study evaluated in vivo efficacy on the system previously described, administering chitosan gels with 2% w/v of atorvastatin to rats with periodontitis induced by ligation. The research found no difference between the water-soluble and basic chitosan formulations in relation to the anti-inflammatory and bone repair activity [188].

## 5. Conclusions

The group of green products is wild and huge and among these, green cosmetics are increasing their marketing power [189]. This review was mainly focused on the physicochemical and biological properties of chitin and chitosan as marine-based natural polymers, and their potential use as starting material for cosmetics. These polysaccharides and their respective derivatives have gained much attention in many cosmetic formulations due to their high percentage of nitrogen (6.8%), their structural characteristics (MW, DD, viscosity, and solubility) and their peculiar biological properties as antibacterial and antioxidant potential agents. One of the major applications of chitin and chitosan is to act as a promising delivery vehicle for active ingredients such as natural compounds, but also as a drug-like active ingredient. In addition, these polysaccharides possess positive charges under physiological conditions that favor the arrangement of a stable system by exploiting the negatively charged nature of the skin, conferring an electrostatic durable interaction. With this review, we aimed to offer an overview of recent developments regarding the use of chitin and chitosan in cosmetic science and its applicability in hair, skin, nails, and oral care, also proposing the use of products containing drugs (cosmeceutical) as an alternative to the classic cosmetics, hoping for upcoming specific regulations.

## Figures and Tables

**Figure 1 marinedrugs-17-00369-f001:**
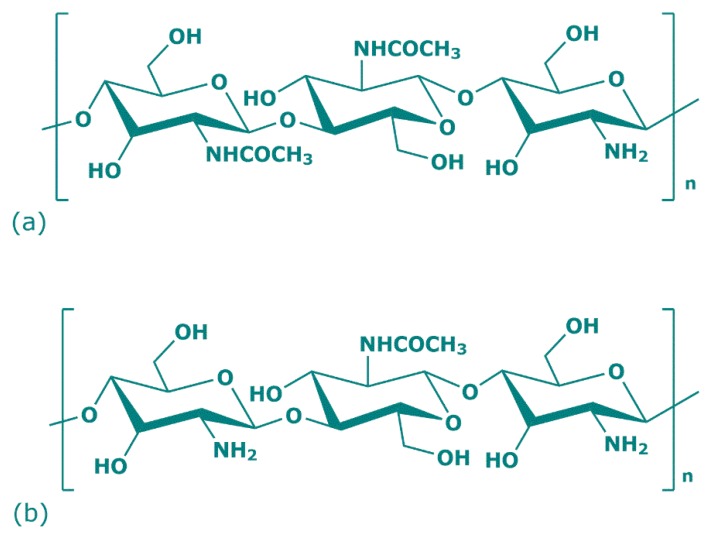
Chemical structure of chitin (**a**) and chitosan (**b**) repeat units.

**Figure 2 marinedrugs-17-00369-f002:**
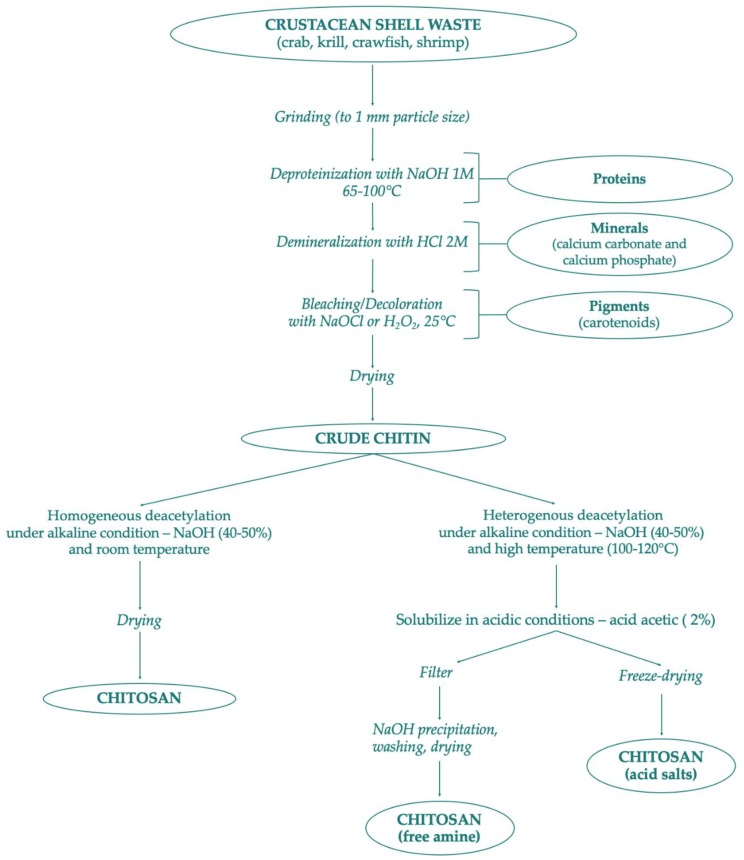
Scheme of chitin and chitosan production following chemical methods.

**Figure 3 marinedrugs-17-00369-f003:**
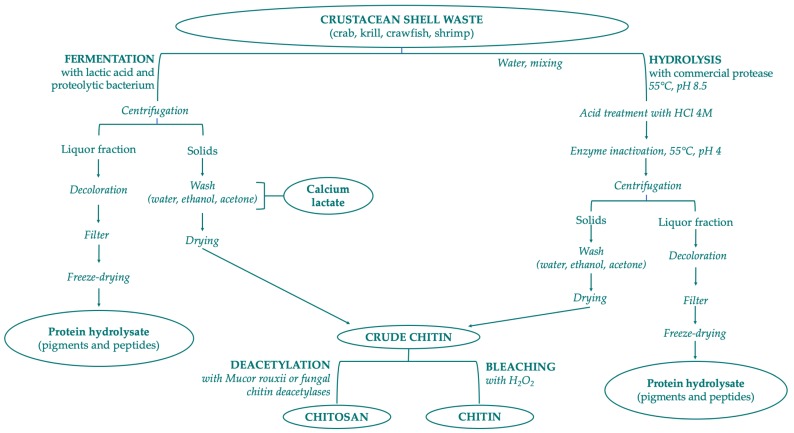
Scheme of chitin and chitosan production following biological methods.

**Table 1 marinedrugs-17-00369-t001:** Influence of DD and MW of polysaccharidic formulation on biological activities.

Effect on Biological Activity	Physicochemical Property
Antimicrobial activity	↑ DD and ↓ MW
Antioxidant activity	↑ DD and ↓ MW
Mucoadhesive properties	↑ DD and ↑ MW
Penetration enhancement properties	↑ DD increased activity, MW is not discriminating

**Table 2 marinedrugs-17-00369-t002:** Chitin and chitosans applications.

	Field	Examples	Ref.
Industrial Applications	Cosmetics	Biodegradable, biocompatible and nontoxic chitosan microparticles encapsulating jabuticaba peel extract	[64]
		Modified chitosan microparticles containing rosmarinic acid for skin delivery formulations	[65]
		Nanoparticles of quaternized cyclodextrin-grafted chitosan associated with hyaluronic acid as promising skin penetration vehicles	[66]
		Preventive effect of chitosan oligosaccharide against UV-caused damage in hairless mouse dorsal skin	[67]
		Periodontal chitosan gels containing moxifloxacin hydrochloride	[68]
		Fluoride loaded chitosan nanoparticles in the prevention of dental caries	[69]
		Hydroxyapatite-chitosan sunscreen antibacterial gel for skin health care	[70]
		Chitosan and surface-deacetylated chitin nanofibrils induced hair growth	[71]
	Agriculture	Gum arabic/chitosan nanoparticles containing geraniol for pest management	[72]
		Chitosan natural biopolymer as a growth stimulator of rice yield	[73]
		Chitosan modified Pt/SiO_2_ as catalyst for an agricultural synergistic agent	[74]
		Antifungal chitosan agent used to control *Ceratocystis fimbriata* plant pathogenic fungus that attacks sweet potato	[75]
		Eco-friendly chitosan/basalt hydrogel as soil conditioner and booster of plants growth	[76]
	Food and Nutrition	Food packaging made by chitosan-based films with microparticles of olive pomace	[77]
		Nisin-loaded chitosan-monomethyl fumaric acid nanoparticles as a direct food additive	[78]
		Chitosan-TiO_2_ nanocomposite film as antimicrobial active food packaging	[79]
		Chitosan as an alternative food preservative to formalin	[80]
		Fish-purified antioxidant peptide-loaded electrospun chitosan/PVA nanofibrous mat for food biopackaging applications	[81]
		Tripolyphosphate and chitosan nanoparticles for encapsulation of C, B9, and B12 vitamins	[82]
		Starch or chitosan-based matrices carrying thyme extract polyphenols as antioxidant films for food preservation	[83]
	Water Engineering – Waste Treatment	Graphene oxide-ionic liquid and magnetic chitosan in heavy metal ion pollution clean-up	[84]
		Multifunctional nanocomposites of chitosan as contaminant water treatment material	[85]
		Antibacterial chitosan chloride-graphene oxide material and/with quartz sand filter media	[86]
	Chromatography	*N*-methoxycarbonyl chitosan for high-performance chiral separation materials	[87]
		*N*-cyclohexylcarbonyl and *N*-hexanoyl chitosans as chiral selectors for enantiomeric separation	[88]
		Chitosan bis(methylphenylcarbamate)-(isobutyrylamide) derivatives as chiral stationary phases for HPLC	[89]
		*O*-carboxymethyl chitosan for convenient use in the purification of lysozyme	[90]
	Paper Industry	Bentonite microparticles/chitosan system for improving the acidic papermaking dry strengths	[91]
		Chitosan/titanium dioxide nanocomposite as antibacterial protective coating for paper packaging	[92]
		Paper wet strength improved with chitosan-based additive using a dipping process	[93]
		Chitosan as antitermite in paper making	[94]
		Caseinate/chitosan films favor reduction in paper water vapor permeability	[95]
	Textile Industry	Series of chitosan-based waterborne polyurethane improve tear strength and antimicrobial activity of polyester cotton dyed and printed fabrics	[96]
		Chitosan and herbal extract of *Aristolochia bracteolate* as medical textile product (band aid)	[97]
		Eco-friendly antimicrobial chitosan-based water dispersible polyurethanes finishes	[98]
		Chitin nanofibers for antibacterial finishing application	[99]
	Batteries	Chitosan networks crosslinked with citric acid or polymeric carboxylic acids as binders for silicon/graphite composite electrodes in lithium ion batteries	[100]
		Molybdenum disulfide-coated nitrogen-doped mesoporous carbon sphere/sulfur composite cathode and carbon nanotube/chitosan modified separator promoting lithium sulfur batteries	[101]
		Chitosan/epoxidized natural rubber networks by crosslinking as a binder material	[102]
		Highly crystalline lithium titanate nanoparticles with N-doped carbon-coating and chitosan (as carbon and nitrogen source)	[103]
		Chitosan composite carbon material with high specific electrochemical performance of lead-carbon battery	[104]
Biomedical and Pharmaceutical Applications	Tissue Engineering	Injectable carboxymethyl chitosan conjugated with α-cyclodextrin hydrogel complexed with poly(ethylene glycol) (PEG_1000_)	[105]
		Electrospun nanofibrous scaffolds containing poly(ε-caprolactone), chitosan, and polypyrrole for neural tissue engineering	[106]
		Alginate/chitosan hydrogel for transplantation of olfactory ectomesenchymal stem cells for sciatic nerve tissue engineering (rat model)	[107]
		Chitosan–vitamin C–lactic acid composite membrane decorated with glycerol and PEG	[108]
		Graphene oxide and amine-modified graphene oxide incorporated into chitosan-gelatin scaffold by covalent linking	[109]
		Magnesium oxide-poly(ε-caprolactone)-chitosan-based composite nanofiber by the electrospinning technique	[110]
		Scaffolds made with modified hydroxyapatite blended into chitosan-grafted-poly (methyl methacrylate) matrix	[111]
	Wound Healing	Collagen/chitosan gel composite supplemented with a cell-penetrating peptide (oligo-arginine R8) with an antibacterial activity	[112]
		Silver nanoparticles encapsulation into chitosan-based membranes without altering the wound-healing ability	[113]
		Rosuvastatin calcium loaded into chitosan hydrochloride scaffolds based with/without mesenchymal stem cells	[114]
		Phenytoin nanocapsules and nanoemulsions formulated as chitosan hydrogels for cutaneous use in rats	[115]
		Electrospun antibacterial PVA/Chitosan/Starch nanofibrous mats	[116]
		Biocompatible and nontoxic PVA/chitosan/nano zinc oxide hydrogels	[117]
	Ophthalmology	Chitosan-covered calcium phosphate nanoparticles loaded with timolol and lisinopril	[118]
		Topical chitosan-N-acetylcysteine for corneal damage in a rabbit model	[119]
		Chitosan-N-acetylcysteine (Lacrimera^®^) in in patients with moderate to severe dry eye disease	[120]
		Contact lenses made of poly(2-hydroxyethylmethacrylate) containing chitosan nanoparticles as dexamethasone sodium phosphate delivery system	[121]
		Timolol maleate imprinted copolymer of carboxymethyl chitosan-g-hydroxy ethyl methacrylate-g-polyacrylamide incorporated on a poly(2-hydroxyethyl methacrylate) p(HEMA) matrix for glaucoma	[122]
		N-Trimethyl Chitosan Nanoparticles loaded with flurbiprofen-hydroxyl propyl-β-cyclodextrin inclusion complex	[123]
		Layer-by-layer deposition of chitosan and alginate was used to control drug release from ophthalmic lens materials	[124]
	Vaccine	Inactivated avian influenza H5N1 virus vaccine encapsulated in chitosan nanoparticles in broiler chickens	[125]
		Chitosan-coated poly(lactic-co-glycolic acid) (PLGA) microparticles for intranasal vaccine delivery of hepatitis B surface Antigen	[126]
		pH-sensitive microneedle chemically coated with inactivated polio vaccine and N-trimethyl chitosan chloride *via* electrostatic interactions for dermal vaccination in rats	[127]
		Glycol chitosan nanoparticles for mucosal intranasal administration of hepatitis B vaccine	[128]
		Folate-chitosan/ interferon-induced protein-10 gene nanoparticles and DC/tumor fusion vaccine enhanced anti-hepatocellular carcinoma effects in mice	[129]
	Drug Delivery	Chitosan-grafted-dihydrocaffeic acid and oxidized pullulan hydrogels *via* a Schiff base reaction for local doxorubicin delivery	[130]
		2-chloro-N,N-diethylethylamine hydrochloride/chitosan pH-responsive nanoparticles as quercetin delivery system for breast cancer treatment	[131]
		pH-responsive Carboxymethyl chitosan nanoparticles for doxorubicin hydrochloride-controlled release at pH 4.5	[132]
		Injectable visible light-cured glycol chitosan hydrogel incorporating paclitaxel-/β-cyclodextrin inclusion complex for ovarian cancer therapy	[133]
		Methyl methacrylate modified chitosan conjugate by a green method *via* Michael addition in curcumin delivery	[134]
	Gene Delivery	Quaternized chitins vector synthesized *via* eco-friendly process	[135]
		Organosilane-functionalized chitosan nanoparticles as plox plasmid delivery system	[136]
		Chitosan-graft-polyethylenimine (PEI)-PEG gene carrier decorated with arginine-glycine-aspartate/twin-arginine translocation for sustained delivery of NT-3 protein growth factor for neural regeneration	[137]
		Targeting ligand conjugated chitosan–PEI copolymer/siRNA polyplexes for cancer therapy	[138]
		Liposome encapsulated chitosan nanoparticles for enhanced plasmid DNA delivery	[139]

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
