# Peer review of "Chitin and Chitosans: Characteristics, Eco-Friendly Processes, and Applications in Cosmetic Science"

_marinedrugs, 2019, doi:10.3390/md17060369_

Reviewer 1 Report

The manuscript has been revised and improved according to the suggested comments. The manuscript can now be accepted for publication.

Author Response

Reviewer 1

Comments and Suggestions for Authors: The manuscript has been revised and improved according to the suggested comments. The manuscript can now be accepted for publication.

The authors thank Reviewer 1 for her/his careful revision and appreciation of our manuscript.

Reviewer 2 Report

It would be usefull to compare the activity of normal chitin /Chitosan with the same polymers in the micro/Nanosize.However,this may the topic of another paper.

Author Response

Reviewer 2

Comments and Suggestions for Authors: It would be useful to compare the activity of normal chitin/chitosan with the same polymers in the micro/nanosized. However, this may the topic of another paper.

The authors thank Reviewer 2 for her/his careful revision. The authors also gladly welcome the suggestion of the Reviewer 2 and we will take that into careful consideration as a starting point for a future manuscript.

Reviewer 3 Report

This paper presents a broad overview of the physicochemical characteristics of chitin and chitosan, ways of their production and possible cosmetic applications. Due to the extremely widespread scientific work related to these polymers, this review is quite relevant and can be accepted for publication.

However, a few minor changes may be proposed:

1. Figure 3 can be edited to improve the readability of the scheme. In this form, reading of small inscriptions is quite difficult.

2. In line 403 is "deacetylation of" missed? enzymatic deacetylation of chitin

3. From the review of source 182 in lines 569-575, the presence and role of chitosan in highlited pastes are not very clear. Perhaps it is worth highlighting the assumption of the authors of the source 182 "The presence of chitosan as a thickener in anti-erosion toothpaste AmF-NaF-SnCl2 allowed ..." in line 573?

Author Response

Reviewer 3

Comments and Suggestions for Authors: This paper presents a broad overview of the physicochemical characteristics of chitin and chitosan, ways of their production and possible cosmetic applications. Due to the extremely widespread scientific work related to these polymers, this review is quite relevant and can be accepted for publication.

The authors thank the Reviewer 3 for her/his careful revision and appreciation of our manuscript which improved its quality and readability.

Specific Comments:

1. Figure 3 can be edited to improve the readability of the scheme. In this form, reading of small inscriptions is quite difficult.

The authors gladly welcome the comment of the Reviewer and therefore improved the quality of the Figure 3.

2. In line 403 is "deacetylation of" missed? enzymatic deacetylation of chitin.

The authors thank the Reviewer for her/his comment. The authors therefore edited the text and rephrased it as follows: “To overcome these drawbacks in the preparation of chitosan, an innovative enzymatic method that exploits chitin deacetylases has been explored: enzymatic deacetylation of chitin”.

3. From the review of source 182 in lines 569-575, the presence and role of chitosan in highlighted pastes are not very clear. Perhaps it is worth highlighting the assumption of the authors of the source 182 "The presence of chitosan as a thickener in anti-erosion toothpaste AmF-NaF-SnCl2 allowed ..." in line 573?

The authors thank the Reviewer for her/his comment. The authors therefore edited the text and rephrased it as follows: “The presence of chitosan as a thickener in anti-erosion toothpaste AmF-NaF-SnCl2 allowed a better preventive effect only for deciduous teeth while the NaF anti-erosion children toothpaste formulation allows better efficacy for both teeth [182].”

This manuscript is a resubmission of an earlier submission. The following is a list of the peer review reports and author responses from that submission.

Round  1

Reviewer 1 Report

General Comments: The submitted manuscript entitled "Chitin and Chitosans: Characteristics, Eco-Friendly Processes and Applications in Cosmetic Science "  by Casadidio et al., presents a review of the Physicochemical properties, the bioextraction methods and applications in cosmetic science for the chitin and chitosan. The manuscript lacks proper discussion and focus on cosmetic application of chitin/chitosan as mentioned in the title. Additionally, section 2 is repetitive and includes unnecessary applications of chitin/chitosan in different fields. The final section on application is also very limited and as such the readers will not gain any new information from this section. Also, the authors fail to discuss factors like properties of chitin/chitosan necessary for application in each area of cosmetic science. Overall, the manuscript is very limited and does not provide any new information on cosmetic applications of chitin/chitosan in comparison to review articles already published. The manuscript can only be considered for publication after significant amount of revision.

Specific Comments:

1.     As mentioned in the abstract, there is no discussion of biological properties of chitin/chitosan. The authors only discuss their applications.

2.     Sections 2.1, 2.2 and 2.3 discusses the same properties and applications repeatedly. The authors should avoid repeating these sections and combine into one section each for Physical and Chemical characterization of chitosan and Applications.

3.     Section 2.1.1., the methods used for determining the DA should be described with examples of chitosan/chitosan derivatives.

4.     Section 2.1.2. The authors only mention the visometric method for measuring the molecular weight of chitosan. Recent advanced and more reliable techniques like Size exclusion chromatography (using Triple Detection Techniques) have not been mentioned.

5.     Table 2 and 3 combines the applications of chitin/chitosan/chitosan derivatives in all different areas. Since, the title of the review mentions applications in cosmetic science, the authors should restrict themselves to the applications in this area.

6.     Table 3 (Applications for chitin/chitosan) should be presented before Table 2 (Applications for chitosan derivatives).

7.     In the section for Applications in Cosmetics, the authors provide a very limited and general discussion for the use of chitosan and its derivatives. The authors should provide elaborate discussion and specific examples of controlling the bacteria or disease using this biopolymer.

8.     Finally, in the conclusion, the authors fail to provide a discussion on what has been developed and the future possibilities and developments needed for enhancing the applications of this biopolymer in cosmetic science.

Reviewer 2 Report

lines 65-67
So-called "natural cosmetics" are not less irritating comparing to "regular cosmetics". It's a sub-group of cosmetics, which should meet all rules of cosmetic regulations. Statements quoted in lines from 65 to 67 are of marketing origin, not scientifically proved. In fact, e.g. plant-based "natural cosmetics" are potentially a "better" source of photosensitising compounds than regular ones — the same concerns heavy metal level, especially in the case of plant-derived raw materials of unknown origin.

line 68
The same concerns "less cosmetic efficacy" statement. It depends on the range of expected efficacy and activity - in case of, e.g. moisturising, "natural" and "regular" could be comparable.

line 93
Typo - should be "neutral" instead of "natural".

lines 124-127
Please consider a refinement of DA term or the introduction of a "degree of deacetylation" term. If the DA (acetylation degree) of chitin turns over 50%, it becomes insoluble in water and other polar solvents. But if deacetylation degree of chitin turns over 50%, the sentence becomes true.

line 394
Skin surface can't be referred to as "mucosal surface." But as all keratin-based structures, it is predominantly negatively charged.

line 397
Please consider a refinement of the sentence. UV-C rays are a component of solar radiation but are fully absorbed and/or reflected in the top layers of the atmosphere, so don't reach the Earth surface from the Sun.

line 411 and 413
Typos - should be "cationic" instead of "anionic" in the case of chitosan and "anionic" instead of "cationic" in the case of the skin.

line 430
The film can remineralise (typo!) only if it contains bioavailable ions. Chitosan films can't "remineralise" nails. My suggestion is to remove the term "remineralize" as it is only of marketing value. Nail keratin doesn't require added mineral compounds to maintain proper hardness or flexibility.

line 445
Keratin contains a lot of amino acids, not only lysine and cysteine. The sentence is misleading.

line 446
The denaturation of hair proteins isn't the primary cause of damages after rapid drying, oxidation etc.

line 452
The film is formed on the hair surface, not in the cortex ("capillary fibers").